# Early Identification of Crop Type for Smallholder Farming Systems Using Deep Learning on Time-Series Sentinel-2 Imagery

**DOI:** 10.3390/s23041779

**Published:** 2023-02-05

**Authors:** Haseeb Rehman Khan, Zeeshan Gillani, Muhammad Hasan Jamal, Atifa Athar, Muhammad Tayyab Chaudhry, Haoyu Chao, Yong He, Ming Chen

**Affiliations:** 1Department of Computer Science, Lahore Campus, COMSATS University Islamabad, Lahore 54000, Pakistan; 2Department of Bioinformatics, College of Life Sciences, Zhejiang University, Hangzhou 310058, China; 3College of Biosystems Engineering and Food Science, Zhejiang University, Hangzhou 310058, China

**Keywords:** crop-type mapping, Sentinel-2, deep learning, crop classification

## Abstract

Climate change and the COVID-19 pandemic have disrupted the food supply chain across the globe and adversely affected food security. Early estimation of staple crops can assist relevant government agencies to take timely actions for ensuring food security. Reliable crop type maps can play an essential role in monitoring crops, estimating yields, and maintaining smooth food supplies. However, these maps are not available for developing countries until crops have matured and are about to be harvested. The use of remote sensing for accurate crop-type mapping in the first few weeks of sowing remains challenging. Smallholder farming systems and diverse crop types further complicate the challenge. For this study, a ground-based survey is carried out to map fields by recording the coordinates and planted crops in respective fields. The time-series images of the mapped fields are acquired from the Sentinel-2 satellite. A deep learning-based long short-term memory network is used for the accurate mapping of crops at an early growth stage. Results show that staple crops, including rice, wheat, and sugarcane, are classified with 93.77% accuracy as early as the first four weeks of sowing. The proposed method can be applied on a large scale to effectively map crop types for smallholder farms at an early stage, allowing the authorities to plan a seamless availability of food.

## 1. Introduction

According to the United Nations estimates, the world population is expected to reach 8.5 billion by 2030, thus escalating pressure on food security (https://population.un.org/wpp/publications/files/key_findings_wpp_2015.pdf, (accessed on 20 December 2022)). Shrinking arable land, climate change, and natural disasters are also hampering the agricultural sector and threatening food security at the global level [1,2,3,4]. The widening gap between food production and its demands is adversely affecting the food supply chain. However, timely and accurate information about cultivated crops and their estimated yields can be incorporated into decision-making to ensure food security. To maintain a seamless food supply chain, remote sensing (RS) data and deep learning (DL) algorithms are crucial in deriving high-quality information on crop-type maps, crop health, and their yields [5,6]. RS is widely used in precision agricultural (PA) applications as it is considered a reliable source for extracting phenological information about crops [7,8,9,10]. The availability of high spectral, spatial, and temporal RS data, including multi-spectral [11,12], hyperspectral [13], and synthetic aperture radar (SAR) [6,14,15,16,17] have opened new possibilities in crop-type mapping [3,18,19,20,21], crop health [22] and yield estimation [4,23,24,25]. In the early 21st century, Landsat and moderate resolution imaging spectroradiometer (MODIS) multi-spectral data were relied on for crop types [26]. However, their capability was limited due to lower spatial and temporal resolution, especially for small fragmented crop fields [27,28]. Nowadays, high-resolution satellite data from the European Union’s Earth Observation Programme and commercially available platforms have unlocked numerous opportunities for crops analysis and monitoring strategies. Table 1 presents some satellite platforms along with their specifications and applications in the agricultural sector. Intelligent algorithms in conjunction with the high-quality remotely sensed data can accurately identify/map the crops and estimate yields. Recently, DL algorithms have proved their significance in remote sensing applications as these algorithms are made to learn from multi-spectral and multi-temporal data of the crops and have outperformed classical machine learning algorithms [29].

Crop identification and mapping using RS-based approaches are more accurate than traditional ground-based methods, which are resource-extensive as the crop information is incorporated at multiple growing stages due to the temporal nature of RS data [40,41]. Optical and microwave remote sensing have both been employed as data sources for PA. Optical RS is based on vegetation indices that are extracted from multi-spectral images. Some of the most often used indices are normalized difference vegetation index, green normalized difference vegetation index, enhanced vegetation index, and land surface water Index [18]. The direct use of time-series vegetation indicators is advantageous for crops with certain temporal characteristics [42]. On the other hand, there has been a considerable rise in using DL algorithms in agriculture applications [43]. The use of deep CNN with multi-spectral and temporal satellite images was pioneered by [44] for rice -crop mapping. Later, many studies used variations of CNN for PA tasks [45,46,47,48,49,50,51,52]. In these patch-based studies, a large area of the land is extracted as a patch and thus requires further processing to extract relevant map of the crops. Additionally, actual field boundaries are not considered in mapping crops as they may have irregular shapes, thus incorporating neighboring fields with different crop types. In Asia, particularly South Asia and China, a smallholder agricultural landscape is prevalent [27,53] where the size of the fields is smaller than 5 ha [54] and crops are diverse. The heterogeneous landscape of small but diverse crops makes it challenging to accurately classify the crops [55,56,57] at an early growth stage. Existing studies on crop mapping and identification approaches mainly focus on large-scale plantation fields (10 hectares on average) and require an entire plant cycle for crop identification [2,9,11]. This study proposes an approach that serves two purposes. Firstly, it focuses on small farm holding (1 to 4 acres on average) and secondly, detects crops at the early stage, by overcoming challenges of overlapping readings due to the plantation of diverse crops in a small region. To the best of our knowledge, no existing study focuses on small farm holding for early crop classification. The objective of this study is to (1) map fields using GPS-based devices to draw field boundaries and acquire crop data, (2) acquire multi-spectral time-series RS data of the mapped fields from Sentinel-2 mission satellites at regular intervals, (3) extract and evaluate basic Sentinel-2 sensors and generate vegetation indices at pixel-level by combinations of bands to find an optimal combination, and (4) apply deep learning model to propose a method to accurately identify staple crops (rice, sugarcane, and wheat) in first 4 weeks of their sowing in smallhold farming, which will allow the government agencies to manage food security effectively.

## 2. Materials and Methods

### 2.1. Study Area

The research is conducted on different districts located in the province of Punjab, Pakistan, the map of which is shown in Figure 1. As per the 2017 census, the total population of Punjab is estimated to be 110 million [58] and it has an area of 20.63 million hectares [59]. Punjab is mostly comprised of flat land and has four seasons which enable suitable conditions to grow staple foods throughout the province. Smallholder agriculture is predominant in the region to provide a livelihood to the residents. Eighty percent of Punjab’s total area is cropped and thus contributes the largest share of the country’s agricultural production. The major crops are wheat, cotton, rice, fodder, maize, and sugarcane.

A ground-based survey was carried out to determine the GPS coordinates of target fields, and other important attributes such as crop history, yields, and sowing dates. During the survey, the coordinates of the field boundaries were recorded using GPS devices, thus giving us the actual field boundaries to acquire satellite data. Fields of varying sizes were mapped at three different sites in the province. The objective of mapping land parcels at three separate sites was to make the dataset diverse and generalized. In total, around 2600 fields were mapped with an average size of 1.2 acres per field. The total mapped area was roughly 4160 acres. To make the dataset relevant and accurate, only the fields were mapped that had been planted with rice, wheat, and sugarcane crops repeatedly for the past three years. The geometry of some of the mapped fields is shown in Figure 1 highlighting the diversity in sizes and shapes of the fields.

The locations of the fields were selected from the Punjab province because it is the agriculture hub of Pakistan and is further divided into three regions. Due to the large size of the province, there is diversity in water patterns, rainfall, and climate. These staple crops are cultivated in all of these three regions, thus, making our dataset represent the entire region.

### 2.2. Sentinel-2 Data Acquisition

The Sentinel-2 mission is a constellation of two satellites named Sentinel-2A and Sentinel-2B [60]. The two satellites are phased at an angle of 180 degrees to one another and are placed in a sun-synchronous orbit around the Earth [61]. These satellites remain stationary in their orbits with respect to the sun (sun-synchronous). The constellation of Sentinel-2 satellites is passive, which means they do not have their own energy source. This is why they are placed in a sun-synchronous orbit to take advantage of the sunlight to capture the earth’s images. To revisit an area of the earth with a single Sentinel-2 satellite, it takes the satellite ten days to return to the same location. However, with the constellation of two satellites, phased at an angle of 180 degrees, it takes them five days to return to the same area of the earth and capture it, thus providing higher temporal resolution [62,63,64].

The Sentinel-2 images are captured with a multi-spectral instrument (MSI) in 13 spectral bands. The multi-spectral instrument takes rows of images as it moves along its orbital path. The light that reaches the multi-spectral sensor of Sentinel-2 is collected by two focal plane assemblies which in turn sample 13 spectral bands. These bands are sampled in visible, near-infrared, and shortwave infrared regions of the electromagnetic spectrum. The 13 bands vary in spatial resolution as four bands are at a 10 meter resolution per pixel, six bands are at 20 meters, and three bands are at a 60 meter spatial resolution.

The Sentinel-2 tiles can be acquired as Level-1C or Level-2A imagery [65]. Level-1C Imagery provides “Top of Atmosphere (TOA)” reflectance, which results in a blur image due to the atmospheric reflectance. Level-2A images are atmospherically corrected and named “Bottom of Atmosphere Reflectance (BOA)”. In Level-2A images, the effect of atmospheric reflectance is removed with the help of a band contained in the multi-spectral image. Subsequently, for each of the three sites considered for this study, their respective fields’ area of interest (AOI) was extracted from tiles of the Sentinel-2 satellites of Level-2A, over rice, wheat, and sugarcane crops, throughout the season.

The period of the acquisition of Sentinel-2 tiles was determined depending on the rice, wheat, and sugarcane crop seasons. In Punjab, the rice crop season begins in June and lasts until October, whereas the wheat crop season begins in December and is harvested in April. However, the sugarcane crop season spans the entire year, with its plantation starting in March. The duration and number of tiles acquired for each crop are detailed in Table 2.

For each map field, 30 multi-spectral images were captured, at five-day intervals, for all three crops. Consequently, the dataset has 30 images of each field for the entire rice and wheat season. However, for the sugarcane crop, the data over five months was acquired to match the temporal length of the other two crops which proved to be sufficient for the sugarcane field classification. During the crop growth cycle, weather conditions have a significant impact on the pattern of crop growth and its yield. To account for the effect of weather conditions, and make the proposed strategy more generalized, the data was acquired for the years 2019, 2020 and 2021. As discussed in Section 2.1, only those land parcels were mapped for the study where the target crops are repeatedly planted for the past three years. This led us to inculcate the impact of soil, weather, and atmospheric conditions in the dataset.

### 2.3. Fields Stripping and Preprocessing

During the crop cycle, we acquired Sentinel-2 tiles for three mapped areas at regular intervals, where each tile contained all mapped fields at the corresponding site. As a single tile covers a large area of the earth’s surface, areas outside the mapped fields were discarded by filtering out the pixels of the tiles that lie outside the boundaries of the mapped fields. This was achieved by creating a mask over mapped fields using fields coordinates and iteratively applying it to the tile. A true color tile of a site and stripped fields are represented in Figure 2.

Cloudy weather causes the multi-spectral instrument of Sentinel-2 to capture the reflectance of clouds. As a result, the true reflectance of the land’s surface is not captured, and the tile is characterized by foggy pixels [66]. Foggy pixels have very high reflectance relative to the land’s surface reflectance. This alters the true reflectance of the surface and causes noise in the data. The reflectivity of the land’s surface is also affected by the shadows of the clouds on the ground. To cope with this problem, spectral band 10 contains information about cloudy pixels. By using band 10, a cloud mask was applied on all tiles to filter out cloudy pixels and replaced them with the average of the adjacent pixels. Further, to make all the Sentinel-2 bands spatially uniform, these were resampled to 10m spatial resolution using “Geospatial Data Abstraction Library (GDAL)” [67].

### 2.4. Selection of Spectral Bands and Vegetation Indices

#### 2.4.1. Spectral Bands

Based on the literature and considering the relevance of the Sentinel-2 bands for agricultural applications, ten bands were chosen for the crop identification task [68,69]. Band 1, band 9 and band 10 were ignored as they do not capture the reflectance of the land surface. Instead, they are used for atmospheric correction, water vapour measure, and cloud masking, respectively [70]. The bands selected for the study are listed in Table 3 along with their bandwidths and central wavelengths.

#### 2.4.2. Vegetation Indices

Vegetation indices are calculated by combining the surface reflectance of two or more spectral bands to obtain a single value [71,72]. These indices serve as indicators of the many properties of the vegetation under consideration [73]. Some indices are beneficial in identifying the vegetated areas, while others are good at determining the health of the plants. There are many vegetation indices proposed in the literature, however, for this study, “Normalized Difference Vegetation Index”, “Green Normalized Difference Vegetation Index”, “Modified Chlorophyll Absorption Reflectance Index”, “Enhanced Vegetation Index” and “Land Surface Water Index” are selected based on their importance in crop identification. 

##### Normalized Difference Vegetation Index (NDVI)

NDVI is employed to assess the extent of vegetation [74,75] and its value ranges from −1 to 1. Densely vegetated areas have NDVI values close to 1, whereas for sparsely vegetated areas the values are near −1. This value is determined from band 4 (red) and band 8 (near-infrared). They operate on the premise that plants require sunlight for photosynthesis to occur. Plant leaves have a pigment known as chlorophyll that absorbs red light, which is required for photosynthesis and reflects near-infrared light. A healthy plant or a densely vegetated region will absorb more red light and reflect less of it. On the contrary, unhealthy plants reflect more red and less near-infrared light.

The NDVI index was chosen because it is a trait that can be used to distinguish between crops. For example, when compared to wheat, the rice crop turns thickly green after only a few weeks of growth after planting. The trend of the NDVI values during crop growing seasons can be quite useful for crop identification. Equation (1) is used to calculate NDVI.
(1)NDVI=(B8−B4)/B8+B4)


##### Green Normalized Difference Vegetation Index (GNDVI)

The GNDVI is identical to the NDVI except that it uses green light instead of red light [76]. Although the normalized difference vegetation index is typically employed to quantify the area of vegetation, the green-normalized difference vegetation index (GNDVI) is an indicator of photosynthetic activity and is, therefore, more useful in yield forecasting [77]. Equation (Equation 2) is used to calculate GNDVI.
(2)GNDVI=(B8−B3)/B8+B3)

##### Enhanced Vegetation Index (EVI)

The EVI [72] index is also used to quantify the amount of vegetation on a given piece of land [78]. Nevertheless, it differs from the NDVI and the GNDVI as it corrects certain atmospheric reflectance that is not considered by the NDVI and GNDVI. It can also identify and eliminate noise in dense plant canopies, allowing for a more accurate estimation of the greenness as a result of the reduced reflectance from noise [79]. Equation (Equation 3) is used to calculate EVI.
(3)EVI=2.5∗(B8−B4)/((B8+6.0∗B4−7.5∗B2)+1.0)


##### Modified Chlorophyll Absorption Reflectance Index (MCARI)

This index measures the concentration of chlorophyll in plants [80]. Low chlorophyll level indicates the weak health of the vegetation. At different crop growth stages, the pattern of the values of MCARI helps to classify the crops [81]. Equation (Equation 4) is used to calculate MCARI.
(4)MCARI=((B5−B4)−0.2∗(B5−B3))∗(B5/B4)


##### Land Surface Water Index (LSWI)

LSWI [82] can distinguish if there is stagnant water on land by observing the reflectance of the near-infrared and shortwave infrared bands [83]. For the plantation of rice crops, the water remains stagnant in the fields for an extended period. By analyzing the time-series data of rice fields, the model can learn the pattern for rice crops and distinguish it from the wheat and sugarcane crop. Equation (Equation 5) calculates LSWI.
(5)LSWI=((B8−B11)/(B8+B11)


### 2.5. Data Shape

For each pixel of the mapped fields, the values of spectral bands and vegetation indices, listed in Section 2.4.1 and Section 2.4.2, respectively, were computed. In the next step, the values of spectral bands and indices were averaged over all the pixels that belong to a particular field. This process was iterated for all the timestamps of all fields separately. This resulted in one numeric value for each band and vegetation index, for a given field at each timestamp image. Consequently, against each field, we have 15 features (10 values for spectral bands and 5 values for indices). Hence, for a single field, an average shape of the data matrix had the dimensions 30 ∗ 15. The complete flow of data acquisition and preprocessing is shown in Figure 3.

### 2.6. DL Network

LSTM is a deep learning network that is considered as one of the best choices for time-series data and data with long-range dependencies [84]. Feedback connections in LSTM distinguish it from the feedforward network. These feedback connections help the model understand the context and long-range dependencies within data. Feedback connections in LSTM enable it to process the whole sequence of data, instead of considering each data point individually. In this way, the information from the previous data points is retained, which is used with the new data points to learn the pattern or data. These characteristics make LSTM a preferred choice over other networks for sequenced and time-series data. LSTM networks make use of gates that control the incoming information, information that needs to be retained for the future, and the information that leaves out of the network. The input gate, forget gates, and output gates are used to control the flow of information [85,86]. Figure 4 shows the architecture of the LSTM DL network, where $x_{t}$ represents the input vector, and $h_{t-1}$ and $C_{t-1}$ are the output and memory from the previous LSTM block, respectively. $C_{t}$ is the memory from the current block that is passed to the next block and $h_{t}$ is the output from the current block. The non-linearities within a single LSTM block are added through sigmoid (σ) and hyperbolic tangent (tanh) functions.

Three stacked LSTM layers each with 256 blocks and a recurrent dropout of 0.3 were used with the data. The LSTM layers were followed by the linear SoftMax layer with three units, which is equal to the number of classes that needs to be predicted. All the experiments were performed using a learning rate of 0.001 with a batch size of 128. The model was trained using cross-entropy loss and an Adam optimizer. The model was trained for 100 epochs with early stopping and a patience counter of 10 epochs. Of the dataset, 80% was used for training while the remaining 20% was used for testing purposes.

### 2.7. Experimental Design

Initial studies were conducted in a set of three experiments on the entire plant cycle of wheat and rice, and the first 5 months of sugarcane cultivation, to identify the best combination of features for crop classification. Experiment 1 focused on the only visible spectrum (RGB bands), experiment 2 includes additional near-infrared and shortwave along with visible spectrum, and lastly, experiment 3 included vegetative indexes in addition to features used in experiment 2. Finally, in experiment 4, features of experiment 3 were utilized by limiting to 1 month of data from sowing instead of the entire plant life cycle for early crop detection.

## 3. Experimental Results

The LSTM model was used to investigate temporal (timestamp) Sentinel-2 data to identify staple crops for small-sized fields at the early stages of the plant life cycle. Optimizing of structure and hyperparameters of LSTM allowed the detection of staple crops within 1 month of sowing with high accuracy.

### 3.1. Model Training

The PyTorch framework is commonly used in academic research to train machine learning and deep learning models. The LSTM architecture was trained and optimized using the PyTorch framework using the different temporal lengths of data ranging from 1 month to the entire plant life cycle. The Colab pro server with 32 GB of RAM was used for training. The LSTM model was trained on our custom dataset in two phases. In the first stage, temporal information of entire plant cycle, at equal intervals, were fed into the network to evaluate the accuracy and significance of different spectral bands. In the second phase, after identifying important spectral bands combinations, the model was further tuned on those selected bands to enable the classification of staple crops within a month of sowing.

### 3.2. Results

The primary objective of the study was to use Sentinel-2 data to identify staple crops for small-sized fields at the early stages of the plant life cycle. In this regard, various experiments were designed to assess the significance of spectral bands, vegetation indices, and temporal length of the input data to classify agriculture fields. The significance of different combinations of Sentinel-2 spectral bands was investigated to identify the best combination of bands in crop classification for small-sized fields. During these experiments, five months from the plantation, timestamped data of the crops were used as input. After determining the best combination of bands in crop classification from five month plantation, additional experiment was performed with one month’s data from plantation, for the early identification of crop type. Table 4 lists the experiments which were performed along with the classification accuracy of the LSTM model on these experiments.

The training accuracy and loss of the optimized model are shown in Figure 5 and their performance on the entire temporal length of a plant cycle and within 1 month of sowing is shown in Table 4.

The accuracy for experiment 1 was 87.55%, which was the worst amongst the three experiments. This was due to the absence of near-infrared bands and vegetative indexes used in other experiments, as the near-infrared bands are crucial to determining the health of the plant and other important features. The reflectance of the near-infrared band greatly varies for crops. For example, in contrast to the wheat crop, the rice crops become dense and dark green after a few weeks of planting and reflect the maximum near-infrared band whose reflectance pattern then determines the crop. The result suggests that the visible spectrum (red, green, and blue bands) alone is not sufficient for crop type prediction.

For experiment 2, the classification accuracy of 98.08% was achieved. The significant increase in the accuracy was achieved by bands in the infrared region as they play a vital role due to reflectance patterns emitted in different plant cycles. In addition to the visible spectrum, the vegetation indices NDVI, GNDVI, EVI, MCARI, and LSWI were generated from near the infrared spectrum. As the harvesting weeks approach, the rice crop stays relatively green compared to wheat and sugarcane. This results in maximum variability in infrared and other bands’ reflectance, and the model was able to generalize and improve accuracy.

In Experiment 3, all ten bands of Sentinal-2 along with the five vegetation indices were utilized. The accuracy of this experiment stood at 99.76 %, the highest accuracy in all experiments conducted. Although some bands play a more crucial role than others, each band captured information differently from the other band, hence the model learned distinguished features in all bands which resulted in better performance than all other combinations. The confusion metrics for all three experiments are shown in Figure 6A–C.

For early identification of the crops, the same parameters from experiment 3 were used in experiment 4, except only using the first month’s data to check the efficiency of our model for early crop-type classification. As the estimated area of the Punjab province is very large (20.63 million hectares), the sowing dates and patterns can vary up to two weeks. Figure 6D shows the confusion metric of this experiment, showing crop type classification accuracy of 93.77%. The results indicate that similar spectral properties of the crops were distinguishable using timestamp multi-spectral features for the entire crop cycle with high accuracy. These features were also useful for detection at an early stage of the crop cycle with reasonable accuracy.

The outcome of the study is a robust model that can classify crops in small fields and which is tolerant to crop diversity in a small region, and is able to predict crops at early stages. The model may fail in some circumstances due to the unavailability of satellite imagery because of bad weather, which may result in fewer readings in each month. Another challenge is the sowing time variation which can hinder accuracy if the sowing time is delayed for more than three weeks from the standard sowing date, for a given region.

## 4. Discussion

The research demonstrates the efficiency of remote sensing imagery, by using different spectral bands of Sentinel-2 and vegetative indices on crop classification for small-sized farms system in Asian countries, at the early stages of the crop. Existing studies focus primarily on large-scale farms for crop classification, whereas in smallholder farming systems, multiple crops are sowed parallel to each other in the same area, thus making crop classification more challenging. Utilizing the multi-spectral sensor to capture distinguishable features from the crops, Sentinel-2 enables observation of fields at regular intervals using 20 m spatial resolution imagery. Our proposed model was able to use distinguishable features from spectral bands and vegetation indices to detect specific crops as early as the first month, with an acceptable accuracy. Furthermore, the study has revealed that the freely available Sentinel-2 data can be utilized in South Asian countries for early crop type mapping for smallholder farms.

In developed countries with large areas, a single crop is cultivated in larger fields covering an area of many hectares and advanced agricultural practices are adopted to monitor crops growth and health. Secondly, a specific region is allocated to the certain crop to be cultivated in the entire region, thus reducing variance in crop types in a particular region. As a result, the precision agricultural tasks such as crop mapping, identification, and monitoring are much easier. However, for this study, considerably small fields were mapped in the state of Punjab, Pakistan, as the smallholder farming system is prevalent in the region. Further, multiple crops are cultivated within a small region in adjacent fields and traditional agricultural practices are used. Conventional farming techniques, the unavailability of data at early stages, and high variance in crop types make it a harder task to automatically identify the crops for yield forecasting and strategic planning.

Similar studies have focused on large-scale farming systems that have employed hyperspectral imaging, thermal imaging, soil moisture, rainfall estimates, etc. [3,4,6,13,16,17,21,25,87]. These studies are required to fuse data from multiple satellite sources to identify similar crops in a given region. The aim of these studies is to find a large contiguous chunk of similar crop cover, and use additional satellite data for calibration. In contrast, we utilize the multi-spectral data from a single Sentinel 2 satellite by only using 10 bands and additional features (NDVI, GNDVI, EVI, MCARI, and LSWI) that enables us to classify crops with high accuracy in small farm systems.

The approach used in this study yielded considerable results for crop identification at an early stage of crop growth despite the small-sized fields. However, some false positives resulted due to the reflectance of the adjacent fields of crops that were not covered in the scope of the study. Limited by the ground truth data availability, only three years of crop type data was assessed in this study. We expect that future long-term field surveys and advanced agricultural practices will result in many other precision agricultural applications for smallholder farming systems to improve crops health and increase yields to ensure food security.

## 5. Conclusions

Existing studies mostly focus on large-scale farming systems comprising of individual fields of an average size of 5 to 10 hectares, resulting in reasonable-sized satellite images to employ CNN-based feature extraction, leading to higher crop-classification accuracy. However, for small-scale fields, these methods cannot be applied, as the average spectral band resolution is 20 m per pixel. Additionally, the regions are dedicated to a single crop in a large farm system and crop diversity does not affect the spectral band readings for a given field, due to the same crops being located in the neighboring fields. In small farm systems, crop diversity is very common, and due to the small size of fields, the effect of neighboring crops is significant. In this study, we propose a robust LSTM-based model the counters the classification challenges of small field-size with high crop diversity, and can classify staple crops in early stages with high accuracy. We verify the effectiveness of our proposed model and demonstrate that our model performs satisfactorily on smallholder farms. In the future, we plan to reduce the detection time further, and incorporate additional satellite data to utilize soil moisture, rainfall estimates, and hyperspectral data to expand this model for more crops.

## Figures and Tables

**Figure 1 sensors-23-01779-f001:**
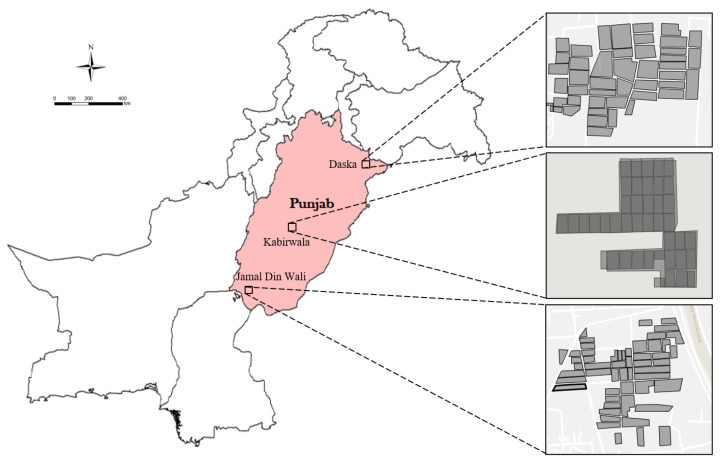
Map of the Province Punjab, Pakistan. The images of the fields of interest are acquired at a resolution ranging from 10m to 60m depending on the bands of Sentinel-2 satellites.

**Figure 2 sensors-23-01779-f002:**
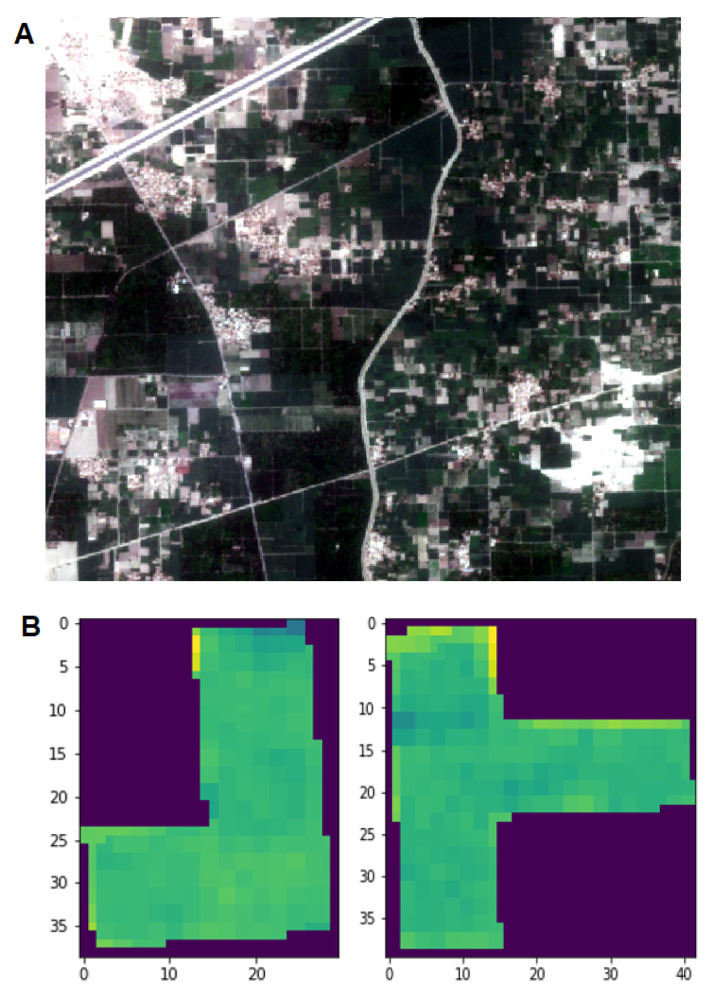
(**A**) A Sentinel-2 tile specimen of sugarcane fields illustrated in RGB bands. (**B**) Depiction of mapped fields obtained by stripping the large tile and visualizing in NDVI values.

**Figure 3 sensors-23-01779-f003:**
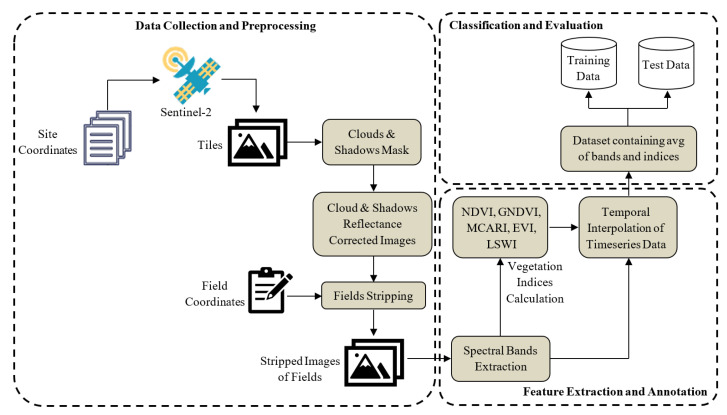
Pipeline for preparation of training and testing datasets used in experimental evaluation.

**Figure 4 sensors-23-01779-f004:**
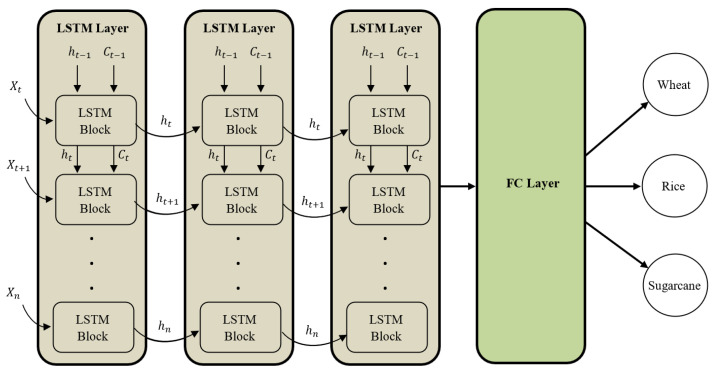
Architecture of the stacked LSTM network.

**Figure 5 sensors-23-01779-f005:**
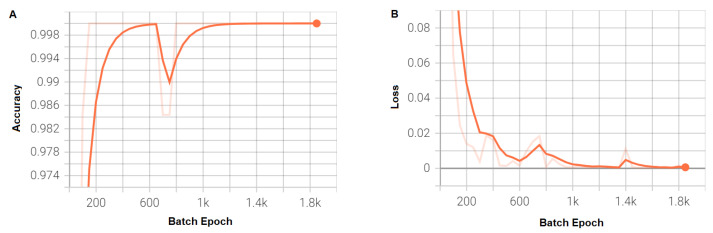
(**A**) Training accuracy. (**B**) Training loss.

**Figure 6 sensors-23-01779-f006:**
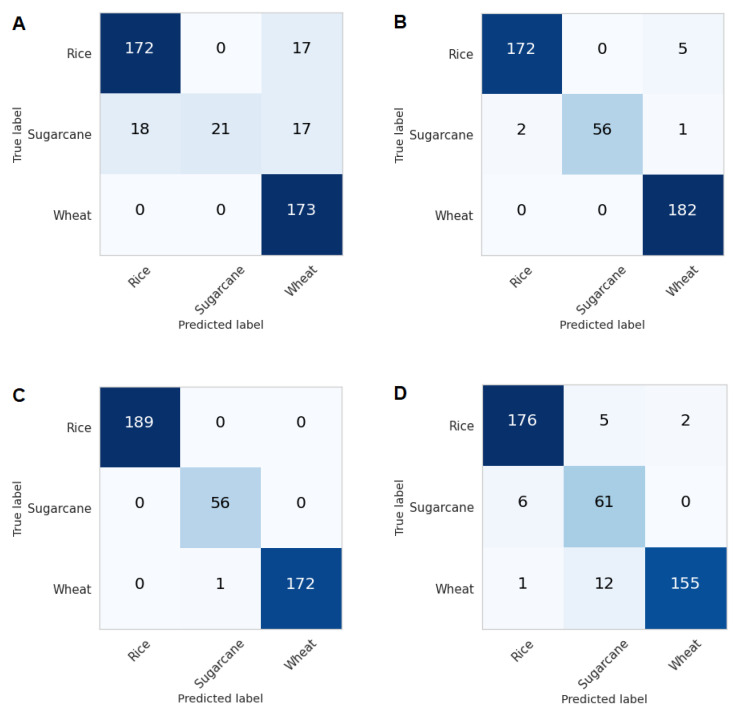
Confusion matrices of (**A**) Experiment 1, (**B**) Experiment 2, (**C**) Experiment 3, and (**D**) Experiment 4.

**Table 1 sensors-23-01779-t001:** Commonly used satellite platforms in agricultural applications along with their specifications.

Satellite	Revisit Time (Days)	Spectral Resolution	Agricultural Applications
Landsat 1 [30]	18	80 m	Crop Growth
Landsat 7 [30,31,32]	16	60 m	Crop Growth, Crop Yield, and Biomass
Landsat 8 [30,31,32]	16	60 m	Crop Growth, Crop Yield, and Biomass
Aqua and Terra MODIS [33,34]	1–2	250–1000 m	Crop Growth and Crop Yield
Rapid Eye [35,36,37]	1–5	6.5 m	Chlorophyll and Water Management
Sentinel-1 [38]	1–3	5–40 m	Crop Growth and Crop Yield
Sentinel-2 [39]	5	10–60 m	Crop Identification and Change Detection

**Table 2 sensors-23-01779-t002:** Testbed Sentinel-2 data.

Crop	No. of Fields	Max Area (Acres)	Mean Area (Acres)	Min Area (Acres)	Total Area (Acres)	Date Range	No. of Tiles
Rice	1200	3	1.2	0.8	1830	1 June–30 October	30
Wheat	1200	3	1.2	0.8	1830	1 December–30 April	
Sugarcane	200	3	1.5	0.7	500	1 March–31 July	

**Table 3 sensors-23-01779-t003:** Description of targeted spectral bands of Sentinel-2 used in experimental assessment.

Band	Bandwidth (nm)	Central Wavelength (nm)
Band 2 (Blue)	65	490
Band 3 (Green)	35	560
Band 4 (Red)	30	665
Band 5 (Red Edge 1)	15	705
Band 6 (Red Edge 2)	15	740
Band 7 (Red Edge 3)	20	783
Band 8 (Near-Infrared)	115	842
Band 8A (Vegetation Red Edge)	20	865
Band 11 (Shortwave Infrared 1)	90	1610
Band 12 (Shortwave Infrared 2)	180	2190

**Table 4 sensors-23-01779-t004:** Experiments with different combinations of spectral bands and vegetation indices and their classification accuracy.

Experiment	Bands	Vegetation Indices	Temporal Length	Accuracy (%)
Experiment 1	Red	NDVI, GNDVI, EVI, MCARI, LSWI	5 Months	87.55
	Green			
	Blue			
Experiment 2	Red			98.08
	Green			
	Blue			
	Near-Infrared			
	Shortwave Infrared 1			
	Shortwave Infrared 2			
Experiment 3	All 10 Bands			99.76
Experiment 4	All 10 Bands		1 Month	93.77

## Data Availability

Freely available data from Sentinel-2 was used. However, coordinates of mapped fields and processed data will be shared upon request.

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
