# Peer review of "Early Identification of Crop Type for Smallholder Farming Systems Using Deep Learning on Time-Series Sentinel-2 Imagery"

_sensors, 2023, doi:10.3390/s23041779_

Round 1
Reviewer 1 Report (Previous Reviewer 2)
The title of this study is: Early Identification of Crop Type for Smallholder Farming Systems using Deep Learning on Time-Series Sentinel-2 Imagery. In this work, the authors wish to (1) acquire multi-spectral time series RS data of mapped fields from Sentinel-2 mission satellites, (2) extract vegetation indices at pixel-level and try these with different combinations of bands to find an optimal one, and (3) apply deep learning model to propose a method to accurately identify the crops (rice, sugarcane, and wheat) in first 4 weeks of their sowing in small hold farming, that will allow the government agencies to manage food security effectively.
I commented on the manuscript and the comments are presented below:
The Introduction to the study is too brief and does not end with a clearly stated purpose. Additional information contained in the Introduction chapter will make the aim of the study will clearly stated. The Methodology section no provides the reader with enough information to repeat the experiments conducted. What were the parameters related to the acquisition of images? How do the techniques used affect the resulting image? These and similar data should be included in the document. Information on the method of statistical analysis of the work results was not included. Image analysis allows to obtain a description of the examined image based on the obtained parameters, e.g. shape, size, color, etc. The obtained parameters can then be subjected to statistical analysis in order to determine the dependence or significance of the obtained results. No statistical analyzes were performed here. For the most parts the 3 and 4 section is well structured. The Discussion is no full comparison and confrontation with the research of other authors in this area. I suggest supplementing the Chapters with additional information related to other new methods and devices in research “Hyperspectral imaging coupled with multivariate analysis and artificial intelligence to the classification of maize kernels”; “Improvement of spatial interpolation of precipitation distribution using cokriging incorporating rain‐gauge and satellite (SMOS) soil moisture data”; “Spatial variability of thermal properties in relation to the application of selected soil-improving cropping systems (SICS) on sandy soil”; “The SMOS-Derived Soil Water EXtent and equivalent layer thickness facilitate determination of soil water resources”; “Real-time global flood estimation using satellite-based precipitation and a coupled land surface and routing model”; “Monitoring agricultural field trafficability using Sentinel-1”; “Coupling a sugarcane crop model with the remotely sensed time series of fIPAR to optimise the yield estimation”; “Assessment of the operational applicability of RADARSAT-1 data for surface soil moisture estimation”; “Potential of sentinel-1 radar data for the assessment of soil and cereal cover parameters”; “Spatio-Temporal mapping of L-Band microwave emission on a heterogeneous area with ELBARA III passive radiometer”.
The Conclusions chapter contains information obtained after conducting experiments but there were no comparison and confrontation with the research of other authors in this area.
Part: References.
The literature used is appropriate but should be supplementing about the items from the last years of publication about similar problem. The literature should be supplemented with additional items describing the examined aspects.
Author Response
Please see the attachment

Reviewer 2 Report (New Reviewer)
In the research presented in the article, the authors took up an interesting topic from a practical point of view. In my opinion, the publication would gain value after taking into account the corrections and additions suggested by me:
- interesting from the point of view of the application of research results, the topic raised by the authors in line 3 was unfortunately not developed in the conclusions and in the discussion part;
- the research results quoted in the Introduction by the authors and included in the literature [1,2,3] are mainly of a local nature and require supplementation;
- it requires an explanation why the authors decided to use the images from the Sentinel-2 satellite in the research, and did not attempt to analyze the images from the Sentinel 1 satellite, which potentially have better quality parameters;
- research objectives described in lines 69-74 largely describe the research methodology, only the objective defined as 3, after modifications meets the conditions of the research objective; it may be worth compiling it with the target specified in lines 264-265;
- in the Materials and methods section, the authors provide two values of the average areas of the analyzed fields: 1.0-1.5 acres (how can you give an average value in the range?!) and 2 acres, which are correct; from the point of view of the resolution of satellite images, it would be worth specifying the ranges of min and max area of fields (Fig. 1 shows that some of the fields are many times larger than others), it would also be worth indicating the location of crops (the shape of the fields in the Kabirwala region indicates the cultivation of rice ( ?)); why were these areas selected as representative for research and analysis?
- in lines 90 the authors write about 4 locations of research, while in figure 1 only 3 locations are visible;
- in line 124, season should be changed to seasones (they are different for different, analyzed crops);
- Table 2 should be supplemented with the total cultivation area of each of the analyzed plants;
- in light of "materials and methods", I don't understand the phrase used by the authors (line 256) "Due to unavailabilty ....
- the terms "experiment 1,..4" contained in table 4 should be previously described in the "materials & methods" section;
- the authors repeatedly use the term "identify crops" in the article, without explaining what they mean by this term (1 - there is, 0 - there is no; what type of crop, what phase of cultivation?)
- how were the results of the "accuracy" value presented in table 4 verified?
Round 2
Reviewer 1 Report (Previous Reviewer 2)
The authors referred to the comments from the previous review for the manuscript titled: Early Identification of Crop Type for Smallholder Farming Systems using Deep Learning on Time-Series Sentinel-2 Imagery. I accept explanations. Authors supplemented the discussion with a new literature data strengthens the message and importance of information in the manuscript.
This manuscript is a resubmission of an earlier submission. The following is a list of the peer review reports and author responses from that submission.
Round 1
Reviewer 1 Report
The paper might contain observations but is not a full study from DL approach. LSTM networks was not illustrated theoretically and technically to support the author results which makes the research not conducted correctly.
Reviewer 2 Report
The title of this study is: Early Identification of Crop Type for Smallholder Farming Systems using Deep learning on Time-Series Sentinel-2 Imagery.
I commented on the manuscript and the comments are presented below:
Part 1: Introduction.
The Introduction to the study is too brief and does not end with a clearly stated purpose or goals that the Authors wish to pursue. This should be changed. I suggest supplementing the Chapter with additional information related to other new methods and devices in studied research.
Part 2: Material and Methods
The Methods section provides the reader with enough information to repeat the experiments conducted but some data is missing. No statistical analysis was used to describe the results. It was not specified what sizes or areas the studied fields occupied. What was the resolution of the photos taken? Was there a clear boundary between the areas? Was it not that there was no clear border and two or more small fields sown with the same plant counted as one field? How was wheat distinguished from other grains?
Part: 3 Results
For the most part, the Results section does not have the appropriate structure to compare the results with the methods used.
Part: 4 Discussion
In the Discussion chapter, there is no full comparison and confrontation with the research of other authors in this area. The results were not fully discussed. A full discussion of the results obtained with other work in this field should be carried out in more aspects. I suggest supplementing the Chapter with additional information. It is difficult to say what was obtained because the manuscript does not have a Conclusions section.
Part: References.
The literature used is appropriate but should be supplementing about the items from the last years of publication about similar problem.